# Anti-Proliferative, Anti-Angiogenic and Safety Profiles of Novel HDAC Inhibitors for the Treatment of Metastatic Castration-Resistant Prostate Cancer

**DOI:** 10.3390/ph14101020

**Published:** 2021-10-04

**Authors:** Zohaib Rana, Sarah Diermeier, Fearghal P. Walsh, Muhammad Hanif, Christian G. Hartinger, Rhonda J. Rosengren

**Affiliations:** 1Department of Pharmacology and Toxicology, University of Otago, Dunedin 9016, New Zealand; ranzo073@student.otago.ac.nz; 2Department of Biochemistry, University of Otago, Dunedin 9016, New Zealand; sarah.diermeier@otago.ac.nz; 3School of Chemical Sciences, University of Auckland, Private Bag 92019, Auckland 1142, New Zealand; fwal135@aucklanduni.ac.nz (F.P.W.); m.hanif@auckland.ac.nz (M.H.); c.hartinger@auckland.ac.nz (C.G.H.)

**Keywords:** HDAC inhibitors, prostate cancer, drug safety, anti-angiogenesis, metallodrugs

## Abstract

Metastatic castration-resistant prostate cancer (CRPC) has a five-year survival rate of 28%. As histone deacetylases (HDACs) are overexpressed in CRPC, the HDAC inhibitor suberoylanilide hydroxamic acid (SAHA) was trialled in CRPC patients but found to be toxic and inefficacious. Previously, we showed that novel HDAC inhibitors (Jazz90 *(N*1-hydroxy-*N*^8^-(4-(pyridine-2-carbothioamido)phenyl)octanediamide) and Jazz167 ([chlorido(η^5^-pentamethylcyclopentadieny[1–4](*N*1-hydroxy-*N*^8^-(4-(pyridine-2-carbothioamido-κ^2^*N,S*)phenyl)octanediamide)rhodium(III)] chloride) had a higher cancer-to-normal-cell selectivity and superior anti-angiogenic effects in CRPC (PC3) cells than SAHA. Thus, this study aimed to further investigate the efficacy and toxicity of these compounds. HUVEC tube formation assays revealed that Jazz90 and Jazz167 significantly reduced meshes and segment lengths in the range of 55–88 and 43–64%, respectively. However, Jazz90 and Jazz167 did not affect the expression of epithelial-to-mesenchymal transitioning markers E-cadherin and vimentin. Jazz90 and Jazz167 significantly inhibited the growth of PC3 and DU145 spheroids and reduced PC3 spheroid branching. Jazz90 and Jazz167 (25, 50 and 75 mg/kg/day orally for 21 days) were non-toxic in male BALB/c mice. The efficacy and safety of these compounds demonstrate their potential for further in vivo studies in CRPC models.

## 1. Introduction

Prostate cancer is the second-most commonly diagnosed cancer in men, and approximately 1.3 million new cases were diagnosed worldwide in 2018 [1]. It also ranked second for cancer-related fatalities in men [2]. Prostate cancers are initially androgen-dependent, and, therefore, androgen ablation therapy is effective. However, the majority of patients progress to a castration-resistant phenotype, which further evolves into an androgen receptor (AR)-negative phenotype [3,4,5]. To date, no treatments are available for AR-negative prostate cancers. Therefore, there is an urgent need for treatments that do not rely on targeting AR. Histone deacetylases (HDACs) are overexpressed in AR*−* and AR+ prostate cancers [6], and, therefore, HDACs might be an important drug target for prostate cancers. Clinically approved HDAC inhibitors (HDACis) such as suberanilohydroxamic acid (SAHA), romidepsin, pracinostat and panobinostat have been trialled. The clinical trials failed in phase II for solid tumours, including in castration-resistant prostate cancer (CRPC), due to toxicity and a lack of efficacy [7]. Of note, these drugs have been approved for hematological malignancies such as peripheral cutaneous T-cell lymphoma and multiple myeloma [8,9,10]. Specifically for CRPC clinical trials, only 2 out of 27 patients treated with SAHA (400 mg/day orally for 3 weeks) showed disease-free progression, lasting for 84 and 135 days [9]. Similarly, disease progression occurred in 22/35, 29/35 and 30/32 patients administered with romidepsin, panobinostat and pracinostat, respectively [11,12,13]. Furthermore, 11 patients had to discontinue SAHA treatment before 6 months due to toxicity, namely weight loss (7/27), anorexia (16/27), fatigue (22/27) or a rise in creatinine (6/27) [9]. On follow-up, 9 patients recovered, suggesting that the toxicity was drug-induced. Similarly, SAHA elicited kidney toxicities (hyperuricaemia, hyperkalaemia and increased creatinine) and liver toxicity (hyperbilirubinaemia) in clinical trials for thyroid carcinoma and peripheral cutaneous T-cell lymphoma [8,14,15]. Pracinostat, romidepsin and panobinostat also elicited kidney, liver and cardiac toxicities in addition to weight loss, nausea, vomiting and anorexia [11,12,13].

The issues of lack of efficacy can be potentially solved by altering the chemical structure of a drug [16]. The substitution or addition of certain moieties can enhance the efficacy of a compound. For example, thioamides have a higher affinity for nucleophiles and electrophiles, in comparison to amides with a weak carbonyl bond [17]. Nocentini et al. also showed that 2,2’-bipyridyl-6-carbothioamide binds to Cu(II) and Fe(II) and, therefore, inhibits ribonucleotide reductase, halting cell division [18]. This pyridine carbothioamide moiety also increases cell permeability [19]. The coordination of metal centres to bioactive molecules such as matrix metalloproteinases and HDACs is another way to increase potency and improve their pharmacological properties [20]. For example, Ye et al. (2013) added a transition metal phenanthroline moiety to SAHA, which increased its potency by 2- to 23-fold in cervical (HeLa), lung (A549 and A549R) and liver (HepG2 and LO2) cancer cells [21]. With the aforementioned evidence, pyridine carbothioamide was functionalised onto SAHA to produce Jazz90 (*N*1-hydroxy-*N*^8^-(4-(pyridine-2-carbothioamido)phenyl)octanediamide), whereas Jazz167 ([chlorido(η^5^-pentamethylcyclopentadienyl)(*N*1-hydroxy-*N*^8^-(4-(pyridine-2- carbothioamido-κ^2^*N,S*)phenyl)octanediamide)rhodium(III)] chloride) was obtained by coordinating a rhodium(pentamethylcyclopentadienyl) moiety to Jazz90 (Figure 1), and the pharmacodynamic profiles of these compounds were the same as for SAHA [22].

We reported that Jazz90 and Jazz167 halted the cell cycle at the G0/G1 phase and exhibited ~2-fold higher selectivity towards PC3 and DU145 cells compared to non-cancerous cells (PNT1A and NIH 3T3). These compounds were also potent HDAC inhibitors with activity superior to SAHA at reducing the expression levels of VEGF-A and VEGFR-2. Jazz90 was 4-fold more potent at inhibiting HDAC6, and Jazz167 was 2-, 4- and 40-fold more potent at inhibiting HDACs 1, 6 and 8 than SAHA [22,23].

This investigation aimed to determine the in vivo safety and tolerability of Jazz90 and Jazz167. Additionally, the compounds were examined for their effect on epithelial-to-mesenchymal transitioning (EMT) and angiogenesis, which are critical to inhibit metastasis. Their effects on 3D spheroids were also tested to predict their capability to penetrate tumors and inhibit growth. EMT and 3D spheroid investigation was carried out on PC3 and DU145 cells, which lack AR-expression and do not require androgen for their growth. Therefore, PC3 and DU145 cells mimic the features of CRPC. All of this information will serve as a predictor of in vivo compound efficacy.

## 2. Results

### 2.1. Compound Effects on Markers of Angiogenesis and EMT

Previously, we reported that Jazz90 and Jazz167 inhibited HDACs, induced the acetylation of histone-3 and had a cytostatic effect on PC3 and DU145 cells [23]. Jazz90 and Jazz167 also inhibited the expression of angiogenic markers such as vascular growth factor-A (VEGF-A) and vascular endothelial growth factor receptor-2 (VEGFR-2) in a 2D model of PC3 cells [22]. In this study, we investigated these compounds further for their effects on pathways such as angiogenesis and EMT that facilitate the dissemination of cancer to other organs. As these compounds inhibited angiogenic markers [22], the compounds were investigated for their ability to modulate human umbilical vein endothelial cell (HUVEC) tube formation. Jazz90 and Jazz167 reduced the number of meshes by 88 and 55%, respectively, and SAHA decreased the number of meshes by 88% (Figure 2e and Appendix A). However, the apparently weaker effect elicited by Jazz167 was not significantly different from the other two compounds. Total segment length was also reduced by 64 and 43% in response to Jazz90 and Jazz167, respectively. Even though SAHA decreased the segment length by 80% (Figure 2f and Appendix A), this also was not significantly different from Jazz90 and Jazz167. To determine if the various anti-angiogenic effects could potentially be driven by the EMT markers, E-cadherin and vimentin, these proteins were assessed in PC3 and DU145 cells by Western blotting. In PC3 cells, E-cadherin protein levels were increased ~40% following treatment with Jazz167 (1 and 4 µM), while vimentin remained at control levels (Appendix A). In DU145 cells, a similar trend was elicited by Jazz90 (4 µM) (Appendix A). However, none of the responses elicited were significantly different from control. 

### 2.2. Effects on Non-histone Mediated Pathways

HDAC inhibitors can modulate changes in non-histone mediated targets in cells. To determine if the anti-angiogenic and cytostatic actions of Jazz90 and Jazz167 were specifically histone mediated, other pathways such as RAS/MAPK and PI3K/Akt signalling were investigated. These pathways play a role in the proliferation, migration and invasiveness of prostate cancer cells [24,25]. In PC3 cells, all compounds increased pAkt/Akt to some degree. However, only Jazz90 (4 µM) was able to significantly increase pAkt/Akt by 182% (Figure 3c, Appendix A). Similarly, Jazz90 was the only compound able to significantly inhibit pErk/Erk (Figure 3d and Appendix A). Both 1 and 4 µM elicited the same mean decrease of 71%. However, the effect elicited by the higher concentration was more consistent. Interestingly, no compound increased the protein levels of pAkt/Akt in DU145 cells (Figure 4, Appendix A). In fact, all mean values for this protein, while not statistically significant, were below control levels. Unexpectedly, bands for pErk and Erk were not detected in DU145 cells. Thus, these results demonstrate that Jazz90 is able to modulate non-histone targets, but only in PC3 cells.

### 2.3. Effects on 3D Spheroids

To further investigate the anti-invasive and growth inhibitory properties of the compounds, their effects on tumour spheroids were examined. Spheroids are an excellent model to use because 3D cultures mimic tumors better than 2D cultures and because they give an indication of the ability of a compound to penetrate into a tumour. Results showed a reduction in PC3 spheroid area by 88 and 89% following treatment with 4 µM of Jazz90 and Jazz167, respectively, whereas SAHA reduced the area by 98% (Figure 5 and Figure 6). Similarly, in DU145 cells, Jazz90 and Jazz167 reduced the area by 89 and 78%, respectively, whereas SAHA reduced it by 88% (Figure 5 and Figure 6). Branching, which indicates the invasiveness of spheroids, was reduced by a similar magnitude in PC3 cells, where Jazz90 and Jazz167 decreased the branching by 97 and 91%, respectively (Figure 5 and Figure 6). A comparable reduction of 88% in the levels of branching was seen in response to SAHA at a concentration of 4 µM. However, no significant differences were seen in branching processes for DU145 cells (Figure 6b).

### 2.4. Safety and Tolerance of Jazz90 and Jazz167

Since Jazz90 and Jazz167 had equivalent and in some cases superior effects compared to SAHA, their safety and tolerability was determined in mice. This evaluation was critical because SAHA has exhibited toxicity in both humans and rodents [8,9,26]. The safety profile of these compounds was examined using oral doses of 25, 50 and 75 mg/kg/day for 21 days [27,28]. The compounds were well-tolerated by the mice, as no significant differences in weight gain were observed over the 21-day period (Figure 7). However, one of the untreated mice died on day 7 for unknown reasons. All of the mice were necropsied on day 22, and organs (heart, spleen, liver, kidney, testes, lungs and brain) were harvested and weighed. There were no significant differences in organ weight between control and treated mice (Table 1). There were also no significant differences in plasma markers (troponin I (TnI), alanine aminotransferase (ALT) and creatinine), assessing cardiac, liver and kidney function, respectively, between treatment groups. Specifically, TnI ranged from 0.01 to 0.26 ng/mL, while creatinine ranged between 0.12 and 2 mg/dL and ALT activity from 14.7 to 76.6 U/L (Figure 8).

## 3. Discussion

Finding new treatments for CRPC that are independent of the AR is critical due to the emergence of AR- cancers following the use of AR-antagonists such as enzalutamide and bicalutamide [5]. Since HDAC inhibitors such as SAHA, panobinostat, pracinostat and romidepsin have failed in CRPC patients due to a lack of efficacy and toxicity, new treatments are urgently needed [7]. We have previously synthesised and characterised two novel HDACis, Jazz90 and Jazz167, which showed an improved selectivity index compared to SAHA (~2-fold) [22] and were superior in terms of inhibiting angiogenic markers (VEGF-A and VEGFR-2) in PC3 cells [22,23]. Therefore, these compounds were examined further for their ability to modulate angiogenesis and invasion. EMT and angiogenesis are particularly relevant to prostate cancer because the five-year survival rate drops from 98 to 29% if the prostate cancer cells metastasise to other organs [1,29]. Furthermore, angiogenic markers such as VEGF-A, VEGFR-2 and microvascular density are associated with a poor prognosis [30,31,32]. Importantly, Jazz90 and Jazz167 inhibited both HUVEC and PC3 cells, unlike SAHA, which only inhibited the HUVEC tube formation. Of note, angiogenesis involves VEGF-A acting as a chemo-attractant and facilitating the migration of endothelial cells towards PC3 cells [33]. This is an important finding because there are no clinically approved prostate cancer drugs that target angiogenesis. Although VEGF-A inhibitors, such as bevacizumab and aflibercept, and VEGFR-2 inhibitors, such as sunitinib, were investigated in prostate cancer patients, they failed because of a lack of efficacy and side-effects [34]. In contrast to the inhibitory action on VEGF and VEGFR-2 by Jazz90 and Jazz167 [22], other HDAC inhibitors such as sodium butyrate and valproic acid (VPA) have shown pro-angiogenic effects. For example, sodium butyrate (0.5-1 mM) increased the levels of VEGFR-2 in HUVEC cells, and VPA (1 mM) increased HUVEC tube formation [35,36]. These differences might be attributed to the selective inhibition of specific HDAC isoenzymes. Specifically, HDAC5 and HDAC7 have anti-angiogenic effects in HUVECs, whereas HDAC6 has pro-angiogenic effects [37,38,39]. Therefore, the selective inhibition of HDAC6 appears to be a desirable property of new HDACis. Various studies have shown that HDAC6 can directly interact with VEGFR-2 and trigger its endocytosis and degradation by a clathrin-dependent mechanism [35,40]. Our previous studies have shown that Jazz90 and Jazz167 were 3- and 4-fold more potent than SAHA at inhibiting HDAC6 [22].

While Jazz90 and Jazz167 have demonstrated anti-angiogenic effects, their mechanism of action is multifaceted, as the compounds had differing effects on proliferation and angiogenesis-regulating pathways, such as RAS/MAPK and PI3K/Akt pathways. However, only one member for each of these signalling pathways was analysed. Therefore, future studies should address if HDAC inhibitors increase phosphatase expression at a transcriptional level or interact with upstream proteins such as epidermal growth factor receptors.

To gain further insight into the mechanism of action of Jazz90 and Jazz167, their ability to modulate the growth and branching of PC3 and DU145 spheroids was evaluated. Of note, no previous studies have investigated the effect of HDAC inhibitors on PC3 and DU145 spheroids. Interestingly, Jazz90, Jazz167 and SAHA all reduced the growth of spheroids by ~90%, and branching by ~95%. Jazz90 and Jazz167 might be more effective at inhibiting spheroids than SAHA as they feature permeability-enhancing chemical moieties, including a thio-group and an aromatic ring [19,41]. The reduction in the number of branches of PC3 spheroids indicates that Jazz90 and Jazz167 may be able to decrease the invasiveness of cancer cells in a 3D environment. However, Jazz90 and Jazz167 did not have an effect on the EMT markers, E-cadherin and vimentin in 2D cultures. However, 3D spheroids and prostate cancer cell monolayer cultures differ in their gene expression. A study by Härmä et al. (2010) demonstrated that 2D cultures of 10 prostate cancer cell lines, including PC3 and DU145, had higher expression levels of genes involved in mitochondrial and ribosomal functions, mRNA processing, DNA synthesis, mitosis and proliferation [42]. In contrast, genes associated with cell–cell adhesion, metastasis and extracellular matrix turnover were upregulated in 3D spheroids [42]. Of note, clinically approved treatment regimens for prostate cancer such as docetaxel, enzalutamide and abiraterone acetate did not have an effect on prostate cancer spheroids [43]. Only bicalutamide (100 µM) inhibited the growth of spheroids by 40% three days after treatment [43]. However, the reason for this may be the fact that the spheroids were grown for 1 day. This is shorter than other 3D spheroid studies, which allow for the growth of spheroids for a minimum of 3 days before treatment [42,44,45]. Therefore, a similar experimental design should be used to compare the clinically approved drugs with Jazz90 and Jazz167 in future studies.

Effective anticancer agents should be selectively cytotoxic/cytostatic towards cancer cells as opposed to normal cells, ultimately exhibiting minimal toxicity. Previously, Jazz90 and Jazz167 were ~2-fold more selective towards PC3 and Du145 cells compared to SAHA and were non-toxic in zebrafish. In order to progress these compounds further, their safety and tolerability were examined in BALB/c mice. Metal-containing compounds such as platinum-based cisplatin and ruthenium complexes such as NAMI-A have elicited cardiac and kidney toxicities [46,47,48]. However, two rhodium-containing compounds (rhodium citrate II and rhodium metalloinserter complexes) were safe following intraperitoneal (1 mg/kg) and intravenous (1.5 mg/kg) administration in mouse models of breast (4T1) and colorectal (HCT116) cancer [49,50]. Therefore, this demonstrated that rhodium-containing compounds may be better tolerated better than platinum or ruthenium-containing compounds. In contrast, in mice, SAHA (50 mg/kg, po and ip) caused spleen lymphoid hyperplasia and reduced spleen weight by 20% [26,51]. Furthermore, in rats, SAHA (50–150 mg/kg, po) reduced body weight gain (5–15%); decreased food consumption; induced lymphoid depletion of the spleen; and decreased the weight of the kidney, liver and thymus by 5.6–15.2% [52]. As HDAC1 and HDAC2 are highly expressed and play critical roles in the normal functions of kidney, liver, thymus and spleen, their inhibition in these organs results in toxicities ([53,54,55]. Jazz90 and Jazz167 are more selective towards HDAC6, which is not as highly expressed as HDAC1 and HDAC2 in kidney, liver, thymus and spleen [22,53,54,55]. In comparison, SAHA is more selective towards HDAC1 and HDAC2 than HDAC6 [7,22]. Therefore, Jazz90 and Jazz167 may be better tolerated than SAHA due to their selective class II HDAC isoenzyme inhibition. Previous studies highlight that SAHA inhibits these HDACs in the nanomolar range. However, based on body weight gain, organ weight and plasma markers (creatinine, TnI, ALT activity), Jazz90 and Jazz167 (25–75 mg/kg, p.o.) did not cause weight loss, or elicit kidney, cardiac or liver toxicities in mice. Other HDAC inhibitors, namely romidepsin, panobinostat and pracinostat, have also shown cardiac and liver toxicities in humans [13,56,57].

## 4. Materials and Methods

### 4.1. Materials

Castration-resistant prostate cancer cell lines (PC3 and DU145 cells) and human umbilical vein endothelial cells (HUVEC) were obtained from American Type Culture Collection (Manassas, VA, USA). Primary antibodies to E-cadherin, vimentin, pErk, Erk, Akt and pAkt (serine-473) were purchased from Cell Signaling Technology (Danvers, MA, USA). Dulbecco’s modified Eagle’s medium (DMEM) nutrient mixture Ham’s F-12, **β**-tubulin, **β**-actin, sodium chloride, sodium orthovanadate, sodium pyrophosphate, sodium azide, nonidet-P40, EGTA, magnesium chloride and Tween-20 were purchased from Sigma-Aldrich (Auckland, New Zealand). Endothelial growth media (EGM) was obtained from Lonza (Morristown, NJ, USA). Acrylamide, bisacrylamide and sodium dodecylsulfate were purchased from Bio-Rad laboratories (Hercules, CA, USA). TrypLE, penicillin, streptomycin and bovine serum albumin (BSA) were obtained from Gibco (Gaithersburg, MD, USA). Sodium hydrogen carbonate was obtained from Applichem (Council Bluffs, IA, USA). Methanol and PVDF membranes were obtained from Merck Millipore (Bayswater, Australia). Tris and sucrose were obtained from BioFroxx (Einhausen, Germany). Complete mini EDTA-free protease inhibitor was purchased from Roche Diagnostics Corporation (Mannheim, Germany). Matrigel matrix was obtained from Corning (Tewksbury, MA, USA). A TnI ELISA kit was purchased from Abcam (Melbourne, Australia). ALT reagent, X-ray film and bicinchoninic acid (BCA) assay was purchased from ThermoFisher (Waltham, MA, USA). The creatinine (serum) colourimetric assay kit was obtained from Cayman Chemical Company (Ann Arbor, MI, USA). Jazz90 and Jazz167 were synthesised, purified and characterized as previously described [22]. The purity of the compounds was > 95% as established by ^1^H NMR spectroscopy and elemental analysis. SAHA was obtained from AK Scientific. 

### 4.2. Cell Maintenance

PC3 and DU145 cells were maintained in 5% DMEM/Ham’s F12 supplemented with 100 units/mL penicillin, 100 units/mL of streptomycin, 2.2 g/L of NaHCO3 and 100 units/mL penicillin. HUVEC cells were grown in EGM supplemented with 2% FBS, hydrocortisone, hEGF, VEGF, hFGF-B, R3-IGF-1, ascorbic acid, heparin and gentamicin/amphotericin-B. All cells were maintained at 37 °C in a humidified atmosphere of 5% CO_2_.

### 4.3. Western Blotting

Samples containing 10 μg of protein from cellular extracts were resolved using SDS-PAGE at 100 V for E-cadherin and vimentin, whereas 40 μg of protein were resolved for Erk, pErk, Akt and pAkt. After the sample ran down to the bottom of the gel, the membrane was removed and transferred into the transfer buffer. A sandwich containing equilibrated fibre pad and blotting paper and an activated membrane was made in cassettes. A voltage of 100 V was set, and the transfer process was carried out for 90 min. The membrane was blocked with BSA blocking buffer 1 ×, followed by a primary antibody incubation (pErk, Erk, pAkt, Akt, E-cadherin and vimentin) overnight, after which the membrane was washed six times with TBST and incubated with secondary antibody for a period of one hour. After six further washes with TBS, chemiluminescent solutions were then added to the membrane, and X-ray films were exposed to the membrane, after which the films were developed. The films were analysed using a BioRad GS710 densitometer (Hercules, CA, USA), and the protein density was calculated as a percentage of β-actin or β-tubulin. Three independent experiments were carried out.

### 4.4. Endothelial Tube Formation

The Matrigel matrix was thawed at 4 °C one day before the tube formation assay. 96-well plates were placed on a flask containing water at 37 °C. A total of 50 μL of Matrigel matrix was added to each of the wells of the 96-well plates. HUVEC cells were washed with PBS, trypsinised with trypsin-EDTA and suspended in 8 mL of EGM-2 media. HUVEC cells (10,000 cells per well) were seeded into each of the wells, followed by the addition of control (0.5% DMSO) and compound treatments (SAHA, Jazz90 and Jazz167) at concentrations of 4 μM. This concentration was chosen as SAHA was tested at a similar concentration in a previous study, allowing for a direct comparison [58]. The plates were incubated at 37 °C at 5% CO_2_ for 18 hours, and photos of endothelial tube formation were taken (Nikon eclipse ti). Three independent experiments were carried out.

### 4.5. Evaluation of Tumour Spheroids

One day before the assay, the Matrigel matrix was stored at 4 °C. On the day of the assay, a 24-well plate was placed on a flask containing water at 37 °C, and 80 μL of Matrigel was pipetted onto the 24-well plate. The plates with Matrigel were incubated at 37 °C for 20 min, and 3500 cells of PC3 or DU145 were added to the wells. The plates were incubated at 37 °C for 3 days, after which the individual wells were treated with compounds (SAHA, Jazz90 or Jazz167) at 4 μM or vehicle control (0.5% DMSO). The concentration chosen was based on previous studies that used a similar concentration of SAHA [44,59,60]. Spheroids were allowed to grow for 1 week, after which 100 spheroids were analysed using an inverted microscope (Nikon eclipse ti). The parameters (longest diameter, shortest diameter, area and branching) were recorded for each of the spheroids. The area of the spheroids correlates with the growth of the spheroid, whereas branching correlates with the invasiveness of the spheroid [42]. Three independent experiments were carried out in triplicate and representative images are shown.

### 4.6. Animal Housing and Care

Male BALB/c mice (5-7 weeks old) were acquired from the Hercus Taeiri Resource Unit (Dunedin, NZ). The animal ethics committee at the University of Otago approved the studies (AUP-20-26). Mice were housed in IVC cages with woodchip bedding in sterile conditions with food (Reliance rodent diet, Dunedin, NZ) and water ad libitum. Mice were housed at a temperature of 21–24 °C on a 12 h light/dark cycle.

### 4.7. Compound Administration

Mice were orally gavaged daily with Jazz90 and Jazz167 for 21 days. Oral administration and the frequency of administration were based on the clinical trials of CRPC patients with SAHA [8,9]. Doses of 25, 50 and 75 mg/kg for Jazz90 and Jazz167 were determined based on past studies, which showed that SAHA was efficacious at 50 mg/kg but toxic in the range of 50–150 mg/kg [51,52]. Vehicle control mice were dosed with 5% DMSO, and another group was left untreated. Mouse weight was recorded daily. Each group contained five animals, except for the untreated group, where one mouse died unexpectedly on day 7. On day 22, the animals were euthanized using CO_2_, and full necropsies were performed.

### 4.8. Organ and Blood Collection

Blood was immediately drawn from the inferior vena cava using a heparinised 20-gauge needle and placed on ice. After PBS perfusion through the heart, major organs (liver, spleen, kidneys, heart, lungs, brain and testes) were harvested and weighed. Blood stored on ice was then centrifuged at 1,700*g* at 4 °C for 5 min. The plasma was then stored at −20 °C, before TnI, ALT and creatinine assays were carried out.

### 4.9. TnI

An increase in the plasma levels of TnI indicates cardiac toxicity. Although SAHA has not induced cardiac toxicity in humans, other HDAC inhibitors such as panobinostat and dacinostat have shown cardiotoxicity [56,61]. Therefore, TnI was monitored following Jazz90 and Jazz167 treatment. The Rat Cardiac Troponin I ELISA kit (abcam#ab246529) was used to measure TnI levels in the plasma. The manufacturer’s instructions were followed to complete the protocol. A Biorad benchmark plus microplate spectrophotometer was used to read the OD at 450 nm.

### 4.10. Assessment of ALT Activity

Plasma ALT activity was used as an indication of hepatotoxicity, as reports suggest that SAHA and another HDAC inhibitor, pracinostat, elicit liver toxicity [13,14]. The plasma levels of ALT were assessed with the Infinity ALT colourimetric assay kit from Fisher Diagnostics using the manufacturer’s instructions.

### 4.11. Creatinine Levels

Plasma creatinine concentration was measured using the creatinine colourimetric assay kit from Cayman Chemical Company (Ann Arbor, MI, USA) according to the supplied protocol. Linear regression conducted on standard creatinine concentrations was used to determine the creatinine concentrations of the samples.

### 4.12. Statistical Analysis

Analyses dependent on time and compound concentrations were evaluated using a two-way analysis of variance (ANOVA) coupled with a Bonferroni’s post-hoc test, where *p* < 0.05 was considered statistically significant. Data were expressed as mean ± S.E.M. The analysis was conducted and graphs were plotted using GraphPad Prism 8 software.

## 5. Conclusions

Jazz90 and Jazz167 exhibited equipotent anti-angiogenic and anti-proliferative activity in PC3 and DU145 spheroids, but they were better-tolerated than SAHA when administered at similar doses to mice. Interestingly, Jazz167, the respective rhodium(pentamethylcyclopentadienyl) complex to Jazz90, did not show higher activity; however, it had the same safety profile as Jazz90. Therefore, further investigation of these compounds could potentially lead to new treatments for CRPC.

## Figures and Tables

**Figure 1 pharmaceuticals-14-01020-f001:**
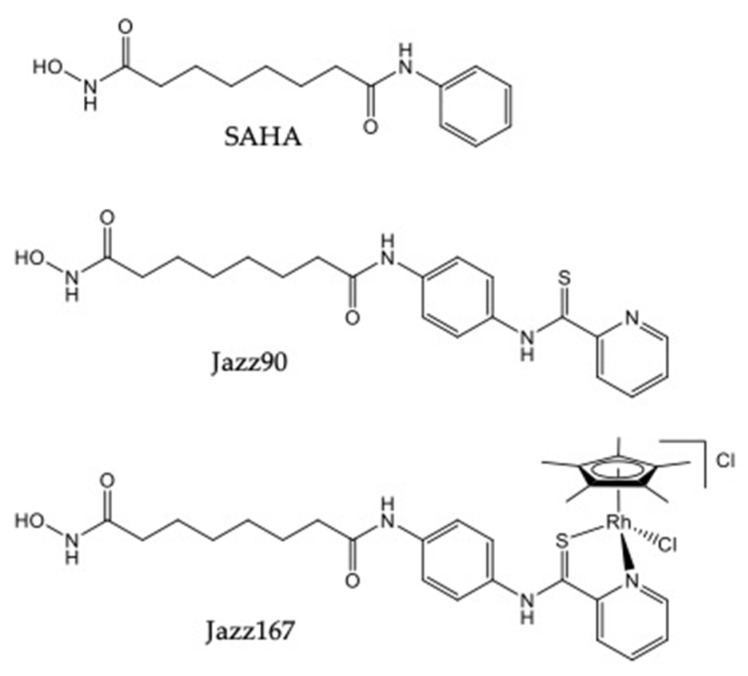
Structures of SAHA, Jazz90 and Jazz167.

**Figure 2 pharmaceuticals-14-01020-f002:**
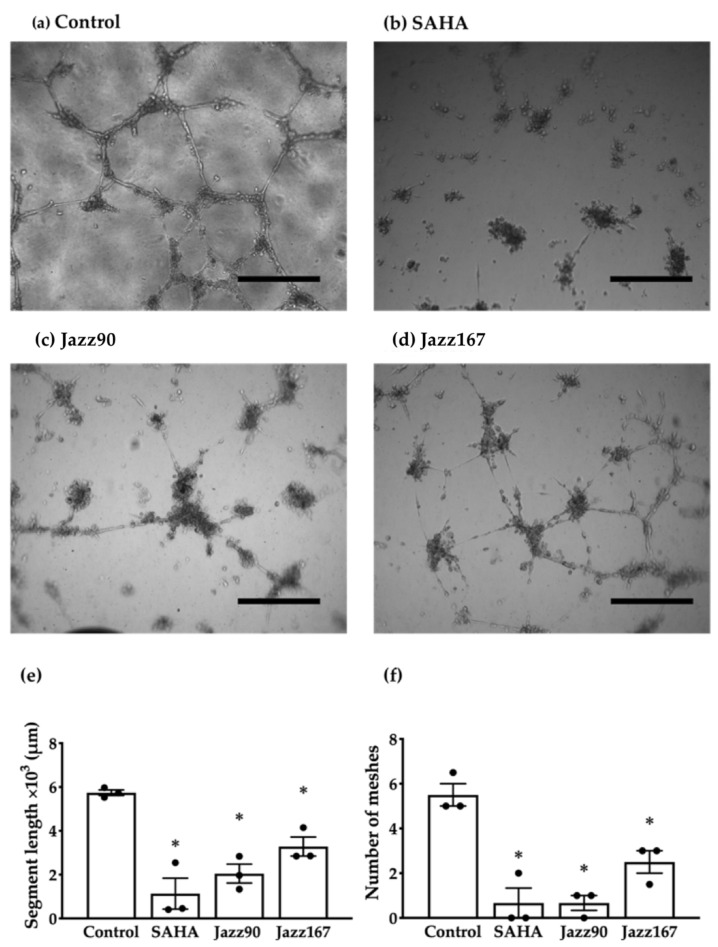
Tube formation in HUVEC cells following HDAC inhibitor treatment. HUVEC cells grown on Matrigel were treated with SAHA, Jazz90 and Jazz167 at a concentration of 4 µM. Control cells were treated with 0.5% DMSO. After 24 h, images were captured using a light microscope. Representative images from HUVEC cells treated with (**a**) control, (**b**) SAHA, (**c**) Jazz90 and (**d**) Jazz167 are shown. The black bars indicate 500 µm at a magnification of ×40. The Angiogenesis analyser plugin was used in ImageJ to quantify the (**e**) number of meshes and (**f**) segment lengths. Bars represent the mean ± S.E.M. from 3 independent experiments, as shown by the data points. Data were analysed using a one-way ANOVA coupled with a Bonferroni’s post-hoc test. *significantly decreased relative to the control, *p* < 0.05.

**Figure 3 pharmaceuticals-14-01020-f003:**
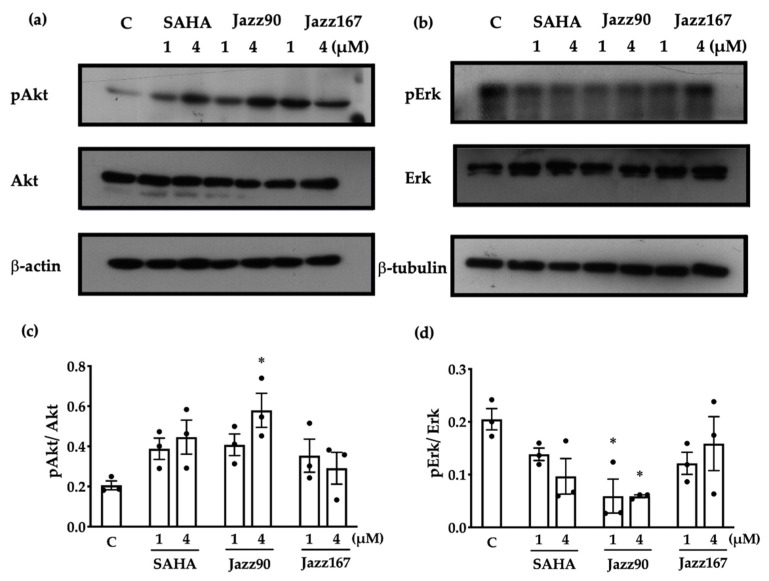
Compound-mediated changes in pAkt/Akt and pErk/Erk in PC3 cells. PC3 cells were treated with SAHA, Jazz90 and Jazz167 at 1 and 4 µM. Vehicle control cells (C) were treated with 0.5% DMSO. Cells were harvested 24 h after treatment. Representative Western blots for (**a**) pAkt, Akt and β-actin (loading control) and (**b**) pErk, Erk and β-tubulin (loading control). Densitometry of scanned Western blots for (**c**) pAkt and (**d**) pErk is shown. Bars represent the mean ± S.E.M. from 3 independent experiments, as shown by the data points. Data were analysed using a two-way ANOVA followed by a Bonferroni’s post-hoc test. *significantly different relative to the control, *p* < 0.05.

**Figure 4 pharmaceuticals-14-01020-f004:**
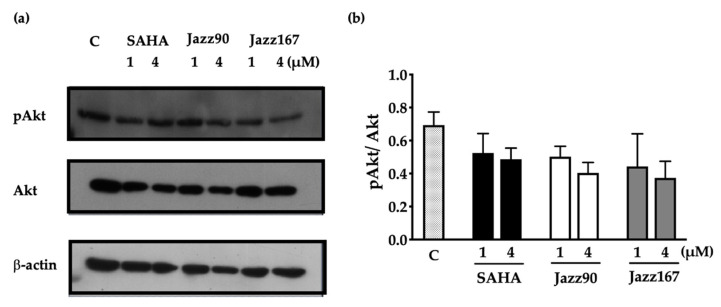
Compound-mediated changes in pAkt/Akt in DU145 cells. DU145 cells were treated with SAHA, Jazz90 and Jazz167 at 1 and 4 µM. Vehicle control cells (C) were treated with 0.5% DMSO. Cells were harvested 24 h after treatment. Representative Western blots of (**a**) pAkt, Akt and β-actin (loading control) and their (**b**) densitometry are shown. Bars represent the mean ± S.E.M. from 3 independent experiments. Data were analysed using a two-way ANOVA followed by a Bonferroni’s post-hoc test. None were significantly different.

**Figure 5 pharmaceuticals-14-01020-f005:**
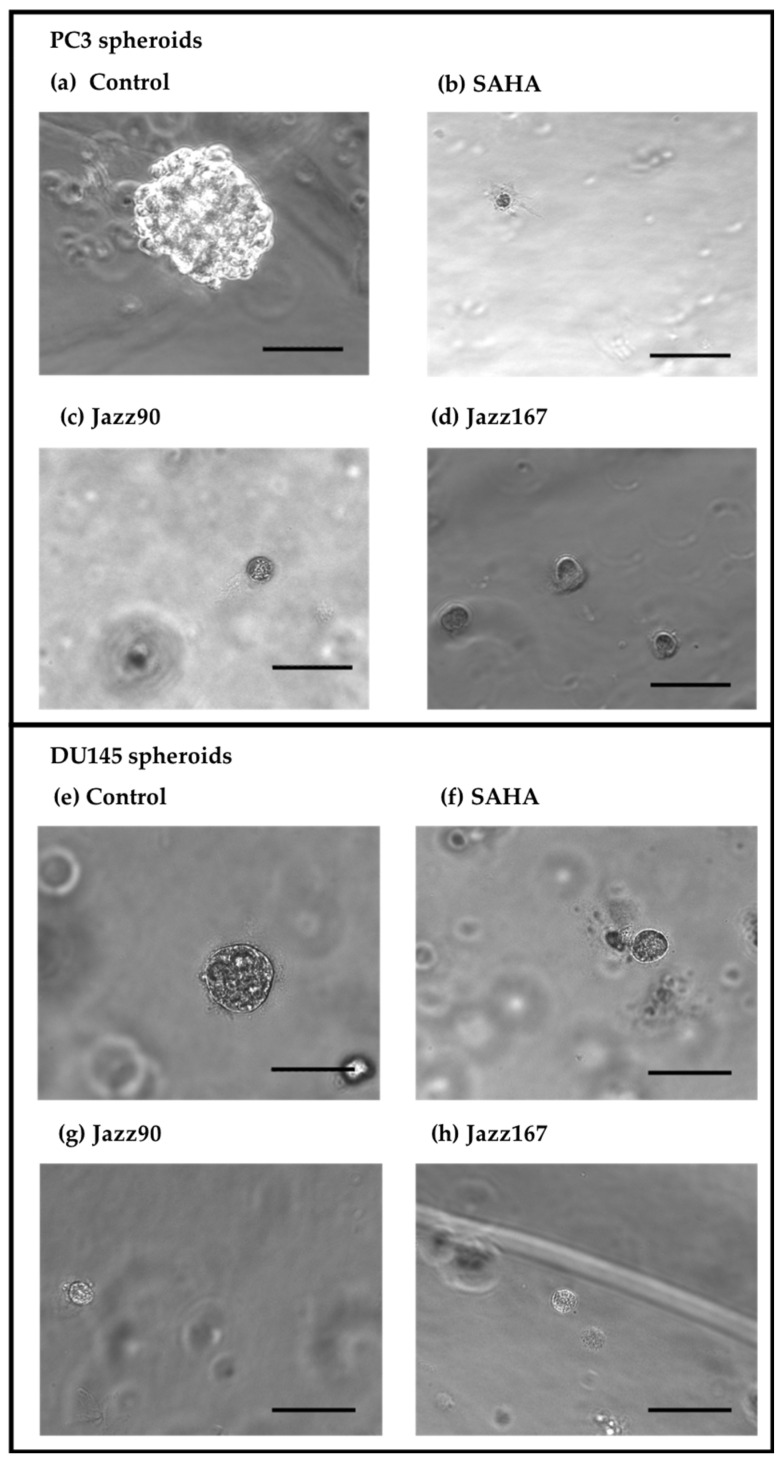
Effect of HDAC inhibitors on PC3 and DU145 spheroids. Jazz90, Jazz167 and SAHA (4 µM) incubated with PC3 and DU145 spheroids grown in Matrigel for one week. Control cells were treated with 0.5% DMSO. Representative images from (**a**,**e**) Control, (**b**,**f**) SAHA, (**c**,**g**) Jazz90 and (**d**,**h**) Jazz167 are shown. The black bars indicate 100 µm at a magnification of × 200. Three independent experiments were carried out.

**Figure 6 pharmaceuticals-14-01020-f006:**
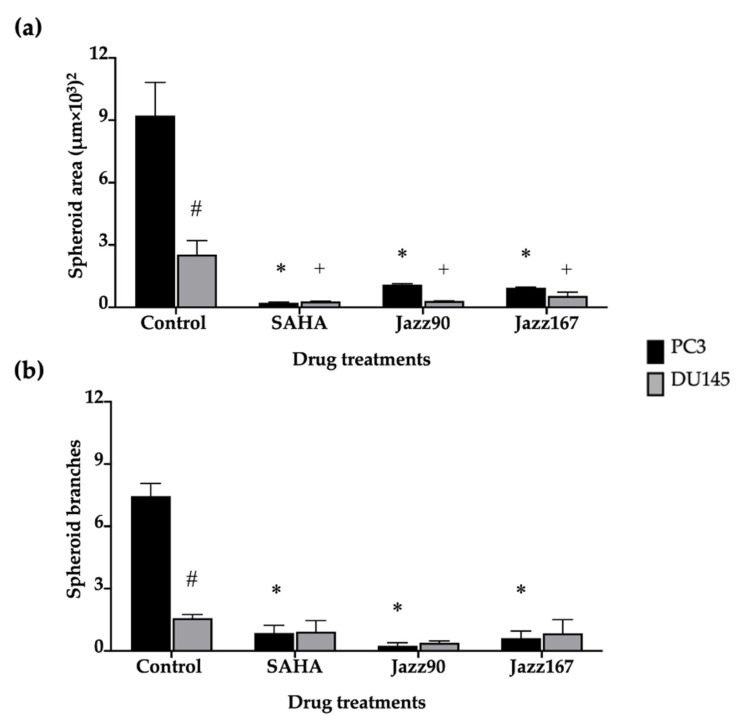
Effect of HDAC inhibitors on the area and branching of prostate cancer spheroids. (**a**) Spheroid areas and (**b**) branches of spheroids formed from PC3 and DU145 cells. Bars represent the mean ± S.E.M. from 3 independent experiments, each containing 100 spheroids. Data were analysed using a one-way ANOVA coupled with a Bonferroni’s post-hoc test. * significantly different compared to control in PC3 cells, #significant reduction in control spheroid areas between PC3 and DU145 cells, + significantly different compared to control in DU145 cells, *p* < 0.05.

**Figure 7 pharmaceuticals-14-01020-f007:**
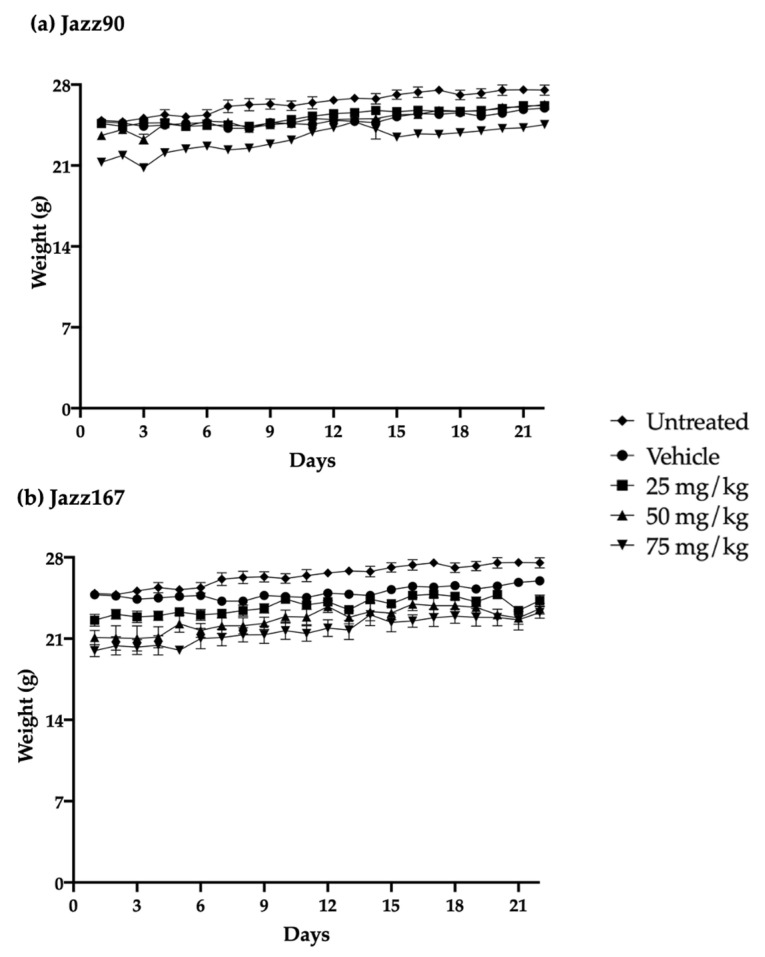
Daily BALB/c mouse body weight throughout Jazz90 and Jazz167 treatment. (**a**) Jazz90 and (**b**) Jazz167 were administered orally for 21 days at 25, 50 and 75 mg/kg. Vehicle control mice were treated with 0.5% DMSO (5 mL/kg). Points represent the mean ± S.E.M from 4-5 mice per group. Data were analysed using a two-way ANOVA coupled with a Bonferroni’s post-hoc test. None were significantly different.

**Figure 8 pharmaceuticals-14-01020-f008:**
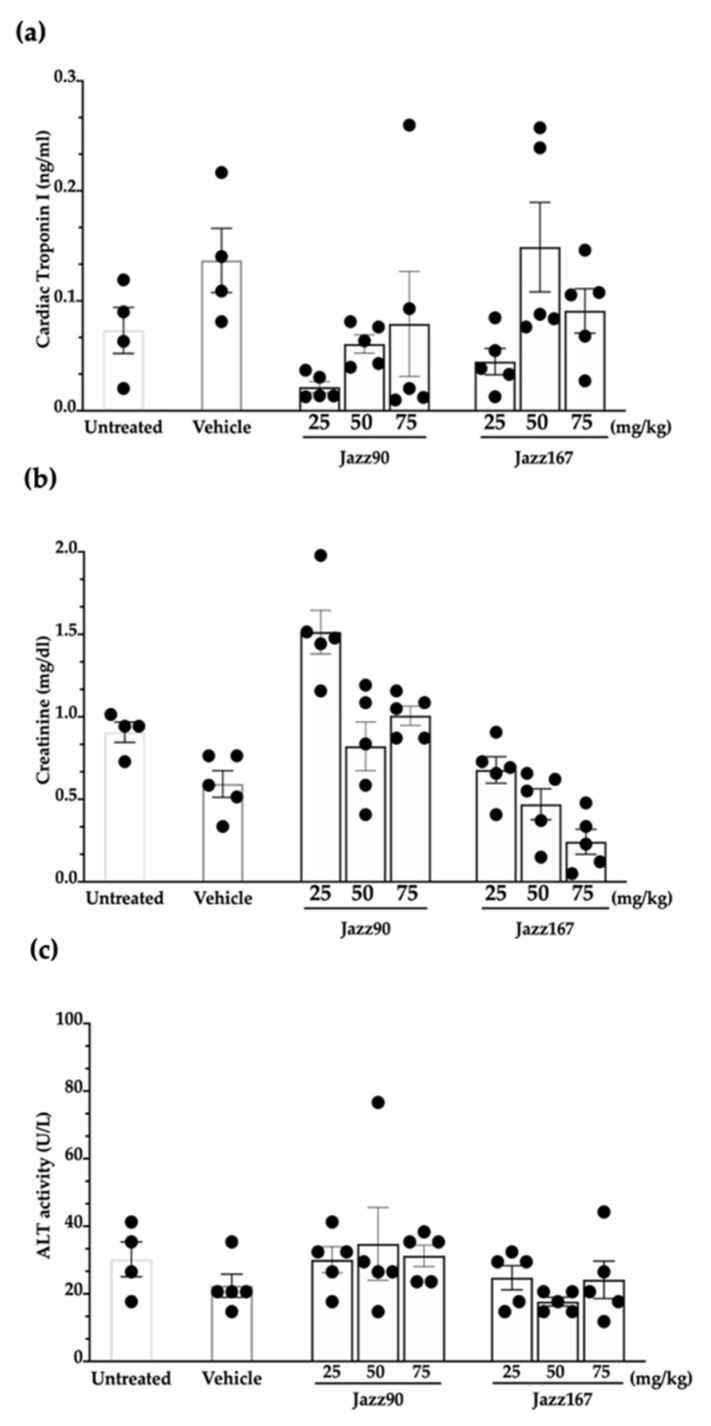
TnI, creatinine and ALT activity in BALB/c mice following compound treatment. Jazz90 and Jazz167 were orally administered at 25, 50 and 75 mg/kg in BALB/c mice daily for 21 days. On day 22, the mice were necropsied, and blood was collected from the inferior vena cava. Plasma (**a**) TnI, (**b**) creatinine and (**c**) ALT activity were measured. Columns indicate the mean ± S.E.M and points represent individual results from each mouse. Data were analysed using a two-way ANOVA coupled with a Bonferroni’s post-hoc test. None were significantly different.

**Table 1 pharmaceuticals-14-01020-t001:** Organ weight (% of body weight) of male BALB/c mice following Jazz90 or Jazz167 treatment.

Treatment	Dose (mg/kg)	Heart	Spleen	Lungs	Liver	Testes	Kidney	Brain
Jazz90	25	0.63 ± 0.03	0.29 ± 0.02	1.22 ± 0.08	5.31 ± 0.2	0.72 ± 0.03	1.83 ± 0.11	1.16 ± 0.06
	50	0.65 ± 0.10	0.30 ± 0.02	1.04 ± 0.17	5.36 ± 0.2	0.65 ± 0.03	1.72 ± 0.10	1.15 ± 0.07
	75	0.66 ± 0.02	0.29 ± 0.03	1.21 ± 0.04	4.93 ± 0.1	0.64 ± 0.05	1.84 ± 0.07	1.25 ± 0.08
Jazz167	25	0.55 ± 0.02	0.28 ± 0.01	0.79 ± 0.07	5.35 ± 0.3	0.69 ± 0.08	1.73 ± 0.07	1.38 ± 0.09
	50	0.55 ± 0.02	0.29 ± 0.02	0.92 ± 0.12	5.26 ± 0.2	0.67 ± 0.01	1.78 ± 0.04	1.39 ± 0.09
	75	0.60 ± 0.05	0.28 ± 0.01	1.11 ± 0.23	5.01 ± 0.2	0.65 ± 0.04	1.72 ± 0.12	1.38 ± 0.09
Vehicle (5% DMSO)		0.56 ± 0.01	0.27 ± 0.01	0.89 ± 0.08	5.10 ± 0.1	0.65 ± 0.02	1.82 ± 0.07	1.39 ± 0.10
Untreated		0.65 ± 0.10	0.30 ± 0.01	1.07 ± 0.15	4.81 ± 0.1	0.68 ± 0.02	1.84 ± 0.03	1.16 ± 0.06

Values are the mean ± S.E.M from 5 mice in each treatment group except untreated, which had an n = 4. Data were analysed using a one-way ANOVA. None were significantly different.

## Data Availability

The data presented in this study are available within the article, associated Supplemental Materials.

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
