# Peer review of "Anti-Proliferative, Anti-Angiogenic and Safety Profiles of Novel HDAC Inhibitors for the Treatment of Metastatic Castration-Resistant Prostate Cancer"

_pharmaceuticals, 2021, doi:10.3390/ph14101020_

Round 1
Reviewer 1 Report
The presented manuscript contains some interesting hints. However some improvements are necessary before the publication.
-the main concern is related to the analysis of the results. In particular, authors in the first part of the experiments compared the results with SAHA, while in the second part of characterization no. I strongly recommend for the experiments reported in Figure 7, Table 1 and Figure 8 to compare the results with a reference drug (i.e SAHA). This is necessary to obtain relevant data for the discussion. Without the comparison with the standard drugs the discussion has not a scientific valence.
minor comments: ml should be mL, a conclusion section should be added reporting the findings of the study
Author Response
Reviewer 1
Query 1: The main concern is related to the analysis of the results. In particular, authors in the first part of the experiments compared the results with SAHA, while in the second part of characterization no. I strongly recommend for the experiments reported in Figure 7, Table 1 and Figure 8 to compare the results with a reference drug (i.e SAHA). This is necessary to obtain relevant data for the discussion. Without the comparison with the standard drugs the discussion has not a scientific valence.
SAHA has been studied in mice and rats at the exact same dose and duration as we used in our experiments. It has also been used extensively in humans. Since its effects in mice have been well-reported we can make our comparison and provide the references as we did in the discussion section. Repeating the experiments for 21 days in mice would have been unethical, as set out by the 3Rs (refinement, reduction and replacement) in animal research policy [1–4]. Therefore, we would not be able to obtain ethical approval to repeat SAHA toxicity experiments from the animal ethics committee at our institution. Additonally, the key information from our experiment was to determine whether or not JAZZ90 and JAZZ167 elecited any toxicity. Adding toxicity results for SAHA, when it has already been published, does not increase or decreas the relevance of the findings.
Query 2: ml should be mL.
This has been changed throughout the manuscript.
Query 3: A conclusion section should be added reporting the findings of the study.
A conclusion section has been added (line 510-515) and now reads as follows.
Jazz9 Jazz90 and Jazz167 exhibited equipotent anti-angiogenic and anti-proliferative activity in PC3 and DU145 spheroids, but they were better-tolerated than SAHA when administered at similar doses to mice. Interestingly, Jazz167, the respective rhodium(pentamethylcyclopentadienyl) complex to Jazz90, did not show higher activity, however, it also had the same safety profile as Jazz90. Therefore, further investigation of these compounds could potentially lead to new treatments for CRPC.
Reviewer 2 Report
Specific comments to the authors
The authors Zohaib Rana et al. of the submitted manuscript „Anti-angiogenic and safety profiles of novel HDAC inhibitors” studied the efficacy and toxicity of the two novel HDAC inhibitors Jazz90 and Jazz167 in-vitro (using the AR- prostate cancer cell lines (PC3 and DU145 cells) and human umbilical vein endothelial cells (HUVEC)) and in-vivo (using Male BALB/c mice).
Based on their investigations the authors could demonstrate that (i) Jazz90 and Jazz167 could significantly inhibit the growth of PC3 and DU145 spheroids and could reduce the PC3 spheroid branching, whereby (ii) Jazz90 and Jazz167 (25, 50 25 and 75 mg/kg/day orally for 21 days) were non-toxic in male BALB/c mice. Therefore, the authors concluded that these findings support the efficacy and safety of these compounds that needs further in-vivo and consecutive clinical evaluation of the clinic-therapeutic potential of these novel HDAC inhibitors Jazz90 and Jazz167 in the future, too.
Overall, the manuscript gives some interesting aspects of efficacy and toxicity of the two novel HDAC inhibitors Jazz90 and Jazz167 in-vitro and in-vivo. The manuscript (including presentation) is comprehensible and convincing. The methods are mostly well described. Although the results and discussion are clear presented, some minor changes must be performed by the authors (see specific comments) to improve the manuscript. In conclusion, the presented data are interesting. After incorporating the mentioned specific comments (see below) the manuscript has the potency to be accepted.
Specific comments
Title: The title is largely unspecific as no relationship is given to (i) the used chemical characteristics of the novel HDAC inhibit, (ii) to the used model of metastatic castration resistant prostate cancer and (iii) to the applied intention/methods (efficacy and toxicity of these compounds).
Abstract: The sentence “Their effects on PC3 and DU145 spheroids were examined to predict their ability to penetrate and inhibit tumor growth.” is not a result and should be transferred or deleted.
Introduction: The conclusion of reference 6 “and, therefore, HDACs are an important drug target for prostate cancers.” is largely optimistic and should be adequately adapted to the clinical reality. Please add “chemical” to “by altering moieties”.
Material and Methods: what is the linkage of the two applied human prostatic cancer cell lines PC3 and DU145 to the topic of “metastatic castration resistant prostate cancer”. Please explain.
Results:
# It is hard to different what was already done and what is new in the experiments in the chapter 2.1. Please clarify.
# Figure 2 and 3: The Western blots are dark for further analyzing.
# Figure 4, 5, 6: These figures could be combined into a common figure.
# Figure 7/8: As the authors stated that “all of the mice were necropsied”, does the histological analysis of the organs reveal any specific changes due to toxicity (especially necrosis or toxic/dysplastic changes of hematopoiesis)?
Discussion: According to the sentence “Of note, clinically approved treatment regimens for prostate cancer such as docetaxel, enzalutamide and abiraterone acetate did not have an effect on prostate cancer spheroids” it would be of interest to compare directly the findings of Jazz90 and Jazz167 with these “classical drugs”. The authors should discuss the limitations of the study (such as analyzing only one member of the RAS/MAPK and PI3K/Akt signaling pathway). Overall, the submitted manuscript with the presented data gives more consecutive question (how is the bioavailability of these drugs? how is the human toxicity? how is the effectivity on different pancreatic cancer cell lines (hormone sensitive/insensitive)?) than giving answers, which must be addressed by the authors. Finally, a definitive conclusion is missing at the end of the discussion, too.
Author Response
Reviewer 2
Specific comments
Title: The title is largely unspecific as no relationship is given to (i) the used chemical characteristics of the novel HDAC inhibit, (ii) to the used model of metastatic castration resistant prostate cancer and (iii) to the applied intention/methods (efficacy and toxicity of these compounds).
The title has been revised and it reads now, “Anti-proliferative, anti-angiogenic and safety profiles of novel HDAC inhibitors for the treatment of metastatic castration resistant prostate cancer”.
Abstract: The sentence “Their effects on PC3 and DU145 spheroids were examined to predict their ability to penetrate and inhibit tumor growth.” is not a result and should be transferred or deleted.
The sentence has been deleted.
Introduction: The conclusion of reference 6 “and, therefore, HDACs are an important drug target for prostate cancers.” is largely optimistic and should be adequately adapted to the clinical reality. Please add “chemical” to “by altering moieties”.
This has been changed and now reads “therefore, HDACs might be an important drug target for prostate cancers.” (line 48)
In response to the second comment the sentence no reads “The issues of lack of efficacy can be potentially solved by altering the chemical structure of a drug” (line 69)
Material and Methods: what is the linkage of the two applied human prostatic cancer cell lines PC3 and DU145 to the topic of “metastatic castration resistant prostate cancer”. Please explain.
This has been addressed at the end of the introduction section (Lines 105-110). This section reads as follows “Their effects on 3D spheroids were also tested to predict their capability to penetrate tumors and inhibit growth. EMT and 3D spheroid investigation was carried out on PC3 and DU145 cells, which lack AR-expression and do not require androgen for their growth. Therefore, PC3 and DU145 cells mimic the features of CRPC. All of this information will serve as a predictor of in vivo compound efficacy.”
Results:
# It is hard to different what was already done and what is new in the experiments in the chapter 2.1. Please clarify.
A section has been added to the beginning of the results to distinguish the previous results more clearly from the studies reporte here (Line 118-125). This now reads as follows “Previously, we reported that Jazz90 and Jazz167 inhibited HDACs, induced acetylation of histone-3 and had a cytostatic effect on PC3 and DU145 cells [23]. Jazz90 and Jazz167 also inhibited the expression of angiogenic markers such as vascular growth factor-A (VEGF-A) and vascular endothelial growth factor receptor-2 (VEGFR-2) in a 2D model of PC3 cells [22]. In this study, we investigated these compounds further for their effects on pathways such as angiogenesis and EMT that facilitate the dissemination of cancer to other organs.”
# Figure 2 and 3: The Western blots are dark for further analyzing.
Densitometeric analysis of the Western blots was conducted by subtracting the dark background. Thus, the analysis was conducted properly. We do not want to do any manipulation to the blot.
# Figure 4, 5, 6: These figures could be combined into a common figure.
The images in Figures 4 and 5 have been combined into one Figure (Figure 5 in the revision). Figure 6 was not combined with the images because spheroid images would become too small to visualize.
# Figure 7/8: As the authors stated that “all of the mice were necropsied”, does the histological analysis of the organs reveal any specific changes due to toxicity (especially necrosis or toxic/dysplastic changes of hematopoiesis)?
We used organ weight and known plama markers to indicate whether or not liver, kidney or cardiac toxicity occurred. These analyses were perform in each individual mouse. Since these results did not indicate toxicity in response to drug treatments, histological analyses for organs were not carried out. This approached was explained in the manuscript and was also supported by the literature.
Discussion: According to the sentence “Of note, clinically approved treatment regimens for prostate cancer such as docetaxel, enzalutamide and abiraterone acetate did not have an effect on prostate cancer spheroids” it would be of interest to compare directly the findings of Jazz90 and Jazz167 with these “classical drugs”.
The discussion has been modified and now reads “Of note, clinically approved treatment regimens for prostate cancer such as docetaxel, enzalutamide and abiraterone acetate did not have an effect on prostate cancer spheroids [43]. Only bicalutamide (100 µM) inhibited the growth of spheroids by 40% three days after treatment [43]. However, the reason for this may be the fact that the spheroids were grown for 1 day. This is shorter than other 3D spheroid studies, which allow for the growth of spheroids for a minimum of 3 days before treatment [42,44,45]. Therefore, a similar experimental design should be used to compare the clinically approved drugs with Jazz90 and Jazz167 in future studies.” (lines 345-352).
The authors should discuss the limitations of the study (such as analyzing only one member of the RAS/MAPK and PI3K/Akt signaling pathway).
This has been listed as a limitation, and future studies have been suggested to understand the mechanism by which RAS/MAPK and PI3K/Akt molecule phosphorylation might be altered (Lines 318-324). This now reads “While Jazz90 and Jazz167 have demonstrated anti-angiogenic effects, their mechanism of action is multifaceted, as the compounds had differing effects on proliferation and angiogenesis-regulating pathways, such as RAS/MAPK and PI3K/Akt pathways. However, only one member for each of these signalling pathways was analysed. Therefore, future studies should address if HDAC inhibitors increase phosphatase expression at a transcriptional level or interact with upstream proteins such as epidermal growth factor receptors.”
Overall, the submitted manuscript with the presented data gives more consecutive question (how is the bioavailability of these drugs? how is the human toxicity? how is the effectivity on different pancreatic cancer cell lines (hormone sensitive/insensitive)?) than giving answers, which must be addressed by the authors. Finally, a definitive conclusion is missing at the end of the discussion, too.
The discussion section has been modified to addressed several of these issues and new sections include the following.
Various studies have shown that HDAC6 can directly interact with VEGFR-2 and trigger it’s endocytosis and degradation by a clathrin-dependent mechanism [35,40]. Our previous studies have shown that Jazz90 and Jazz167 were 3- and 4-fold more potent than SAHA at inhibiting HDAC6 [22]. (lines 314-317)
Line 318 – 324 and 345-452 shown above
Round 2
Reviewer 1 Report
Authors nicely addressed my concerns. The manuscript can be published in the present form